

# A retrospective analysis of a newly proposed imaging-etiologic classification for acute ischemic stroke with large vascular occlusion based on MRI and pathogenesis

Hao Li[1], Zhaoshuo Li[1], Jinchao Xia[1], Lijun Shen[2], Guangming Duan[1] and Ziliang Wang[1]

[1] Department of Cerebrovascular Disease, Zhengzhou University People's Hospital & Henan Provincial People's Hospital, Zhengzhou, Henan, China
[2] Department of Clinical Medical Research Center, Zhengzhou University People's Hospital & Henan Provincial People's Hospital, Zhengzhou, Henan, China

Corresponding author
Ziliang Wang, wzl731023@163.com

## ABSTRACT

**Background:** Endovascular treatment (EVT) has emerged as the preferred initial therapeutic option for acute ischemic stroke (AIS) with large vascular occlusion (LVO). To facilitate more targeted EVT interventions, we propose a novel imaging-etiologic classification system derived from a comprehensive analysis of preoperative MRI and pathogenesis in AIS patients.

**Methods:** From June 2020 to December 2021, a retrospective analysis was conducted on 184 consecutive AIS patients who underwent preoperative MRI and subsequent EVT at the Henan Provincial Cerebrovascular Hospital Stroke Center. Patients' medical histories were comprehensively reviewed. According to MRI, anterior circulation infarction (ACI) and posterior circulation infarction (PCI) were divided into four groups respectively (A-D & a-d). Three types (1–3) of etiology were identified based on pathogenesis. The types were respectively evaluated by screening test with intra-operative finding of EVT.

**Results:** Our imaging-etiologic classification achieved an overall positive rate of 90.2% (166/184) when compared to the gold standard. The screening test for each type demonstrated excellent validity (Youden's index $\geq$ 0.75) and reliability (Kappa $\geq$ 0.80).

**Conclusion:** The imaging-etiologic classification represents a simple yet comprehensive approach that can be readily applied in the management of AIS with LVO. It can rapidly and effectively locate the vascular occlusion, and reveal the pathogenesis.

## INTRODUCTION

Acute ischemic stroke (AIS) is the most prevalent type of stroke, comprising approximately 69.6–70.8% of all stroke cases in China (*Wang et al., 2017a*, *2017b*). Currently, endovascular treatment (EVT) bolstered by the outcomes of five pivotal randomized controlled trials-MR CLEAN, ESCAPE, EXTEND-IA, SWIFT PRIME and REVASCAT (*Berkhemer et al., 2015*; *Campbell et al., 2015*; *Goyal et al., 2015*; *Jovin et al., 2015*; *Saver et al., 2015*), has emerged as the frontline therapeutic option for AIS patients with large vascular occlusion (LVO). The classification of AIS significantly influences patient management, prognostic assessment, secondary stroke prevention strategies, and stroke research endeavors. Currently, the most popular stroke classification system is the TOAST criteria, which provides a theoretical basis for different treatments. However the system does not specifically cater to EVT. Chinese researchers have attempted to devise typing systems based on digital substraction angiography, yet these classifications are often intricate, hindering their widespread adoption and application. With the extension of therapeutic time window, the rates of AIS patients who can be candidates for emergency EVT will increase significantly (*Albers et al., 2018*; *Casetta et al., 2021*; *Nogueira et al., 2018*). Hence, the ability to swiftly, succinctly, and effectively leverage the preoperative data (MRI, medical history, *etc.*) to identify different types of AIS, guide the EVT and expedite the vascular recanalization time is of great importance. This imperative underpins the inception of our study.

## METHODS

### Study population

From June 2020 to December 2021, a total of 184 patients with AIS who underwent EVT at the Henan Provincial Cerebrovascular Hospital Stroke Center were enrolled in the study. Their preoperative MRI (including DWI and MRA) examinations and medical histories could be reliably accessed. Patients were excluded if their CT/MRI/DSA scans were performed at other hospitals and the data were unavailable. Among the 184 patients, 65 of them had a history of heart diseases, such as atrial fibrillation, atrial flutter, mitral valvular disease; 14 cases had malignancy or were undergoing chemotherapy; one case presented with bacteremia; one case had a history of heavy hormone usage; one case had renal failure and required dialysis; 112 cases had hypertension; 40 cases had diabetes; 14 cases had hyperlipidemia; 63 cases were smokers; and three cases had a history of trauma or head-and-neck tearing pain. The median time from stroke onset to treatment was 4.3 h (range 0.3–48 h). The preoperative NIHSS scores for each patient were greater than six. All patients or their family had signed informed consent for EVT. This study was approved by the Ethics Committee of Henan Provincial People's Hospital (approval number: 2020-108). Patients provided informed consent through a process that was reviewed and approved by the Ethics Committee of Henan Provincial People's Hospital, ensuring that the study was conducted in accordance with the ethical standards outlined in the 1964 Declaration of Helsinki.

Inclusion criteria: 1. AIS with LVO diagnosed by imaging and clinical symptoms; 2. Patients meeting the operation indications had received EVT; 3. The medical history, the preoperative MRI and the intra-operative findings of EVT were reliable and available for study.

Exclusion criteria: 1. AIS without LVO was not included in this study; 2. For the anterior circulation, AIS due to isolated anterior cerebral artery occlusion was usually treated conservatively and was therefore excluded; 3. There was no preoperative MRI examination performed before undergoing EVT, or the MRI was unavailable for collection, or the MRI could not be interpreted due to various reasons (such as head implants or motion artifacts); 4. The data regarding EVT or medical history was lost.

## Imaging-etiologic classification of AIS with LVO

The imaging-etiologic classification of AIS with LVO was based on MRI and pathogenesis, and estimated by comparison with the gold standard intra-operative finding of EVT. All image assessments were performed by two independent neurointerventionists before EVT, and discrepancies were resolved by consensus.

Our classification consists of two main components: imaging subclassification and etiologic subclassification, as detailed in Table 1. The imaging subclassification aims at locating the vascular occlusion or severe stenosis, and is based on preoperative MRI which requires at least two sequences, including DWI and MRA. According to the position of large vascular occlusion, anterior circulation infarction (ACI) and Posterior circulation infarction (PCI) were stratified into four categories respectively (A-D & a-d) (Figs. 1, 2).

**Anterior circulation infarction (ACI) (Fig. 3):**

Type-A is defined as occlusion of the distal middle cerebral artery (MCA, M1), specifically the distal one- third of M1, including the M1 bifurcation and M2; DWI shows that the infarctions locate in the cortex, corona radiata and borderzone area. The MCA or terminal internal carotid artery (ICA) disappears on MRA;

Type-B is defined as occlusion of the mid-proximal M1, covering approximately the proximal two-thirds of M1. DWI shows isolated or combined infarctions in the basal ganglia region (such as the caput nuclei caudati, lenticular nucleus, anterior limb or genu of internal capsule infarctions). The MCA or terminal internal carotid artery (ICA) disappears on MRA;

Type-C is defined as occlusion of the intracranial ICA (bounded by the ophthalmic artery), from the ophthalmic segment to bifurcation of the ICA (*i.e.* C6-C7, based on the ICA classification proposed by Bouthillier). The DWI is characterized by posterior limb of internal capsule (PLIC) infarctions. Another characteristic of DWI is the simultaneous occurrence of infarctions in the territories supplied by the anterior cerebral artery (ACA) and MCA. MRA shows that the ICA or intracranial segment of the ICA, or the ICA combined with the MCA and ACA, disappears.

Type-D is defined as occlusion of the extracranial ICA, from the initial ICA to clinoid segment of the ICA (*i.e.* C1-C5). DWI shows there is no infarction related to the PLIC, which is described as PLIC evasion. MRA shows that the ICA or ICA combined with the MCA/ACA disappears.

**Table 1 Imaging-etiologic classification.**

| | | Type | Definition | Segment | DWI | MRA (Disappear) | Major etiology | Other etiology |
|---|---|---|---|---|---|---|---|---|
| Image | ACI | A | Distal M1 | About distal third of M1 (Including bifurcation and M2) | Infarction in cortex, corona radiata and borderzone area | MCA or terminal ICA + MCA | | |
| | | B | Mid-proximal M1 | About proximal two-thirds of M1 | Basal ganglia infarction | MCA or terminal ICA + MCA | | |
| | | C | Intracranial ICA | C6-C7 | PLIC infarction or infarction in ACA⊠MCA simultaneously | ICA or intracranial ICA or ICA + MCA + ACA | | |
| | | D | Extracranial ICA | C1-C5 | PLIC evasion | ICA or ICA + MCA + ACA | | |
| | PCI | a | Distal BA | Distal BA (Including SCA) | Superior cerebellar peduncle level of pons, midbrain, thalamus, occipital lobe infarction or infarction in SCA | Mid-distal BA | | |
| | | b | Mid-proximal BA | From initial BA to SCA (exclusive) | Pons infarction or infarction in AICA | VBA or junction of VBA | | |
| | | c | Intracranial VA | V4 | Medullary infarction or infarction in PICA | VBA or junction of VBA | | |
| | | d | Extracranial VA | V1-V3 | The same as Type-a or large amounts of infarctions in one side | VBA or mid-distal BA+ partial signal of ipsilateral VA | | |
| Etiology | | 1 | LAA | | | | Risk factors for atherosclerosis | Age ≥ 40 |
| | | 2 | Thrombosis | | Multiple infarctions in bilateral anterior, anterior and posterior circulations | | Cardiogenic embolism | Risk factors for HCS, age < 40 |
| | | 3 | Dissection | | | | Atherosclerosis | Trauma, Iatrogenic, *etc.* |

**Posterior circulation infarction (PCI) (Fig. 4):**

Type-a is defined as occlusion of the distal basilar artery (BA), including the superior cerebellar artery (SCA). DWI shows that acute infarctions occur solely in the superior cerebellar peduncle level of the pons, midbrain, thalamus, occipital lobe and/or the territory supplied by the SCA. MRA reveals that the mid-distal basilar artery disappears;

Type-b is defined as occlusion of the mid-proximal BA, from the origin of the BA up to but excluding the SCA. DWI shows that the infarctions involve the pons (excluding the superior cerebellar peduncle level)or regions supplied by the anterior inferior cerebellar

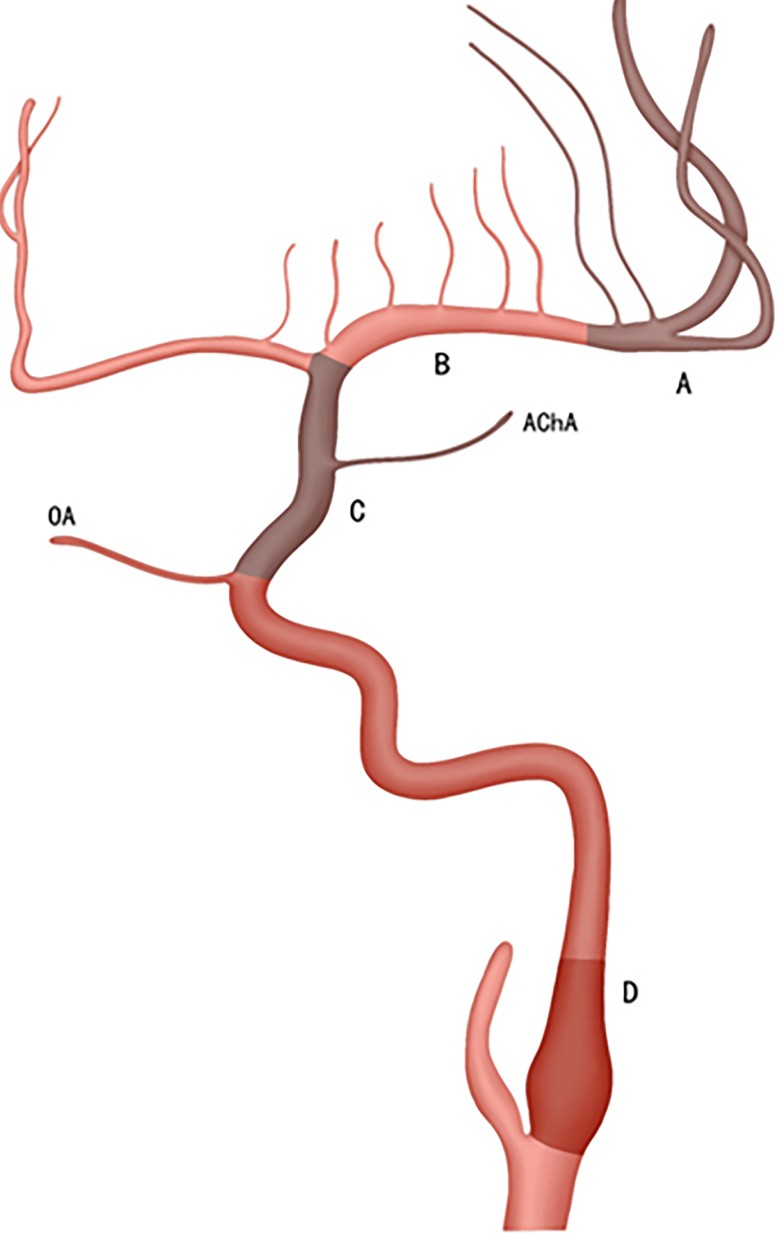

**Figure 1** **The ACI is divided into four types according to the location of vascular occlusion.** A, B, C, and D; Type-A involves the distal one-third of M1; including the bifurcation and M2; Type-B covers the proximal two-thirds of M1; Type-C corresponds to the intracranial segment of ICA (*i.e.*, C6-C7); Type-D encompasses the extracranial segment of the ICA (*i.e.*, C1-C5). Additionally, occlusion of the initial part of the ICA is more frequently observed.

artery (AICA). MRA indicates that the entire segment or the junction of the vertebral-basilar artery (VBA) disappears;

Type-c is defined as occlusion of the intracranial segment of vertebral artery (VA), specifically the fourth segment of VA (*i.e.* V4). DWI shows that the infarctions involve the medulla or the area supplied by the posterior inferior cerebellar artery (PICA). MRA shows the entire segment or the junction of the vertebral-basilar artery (VBA) disappears;

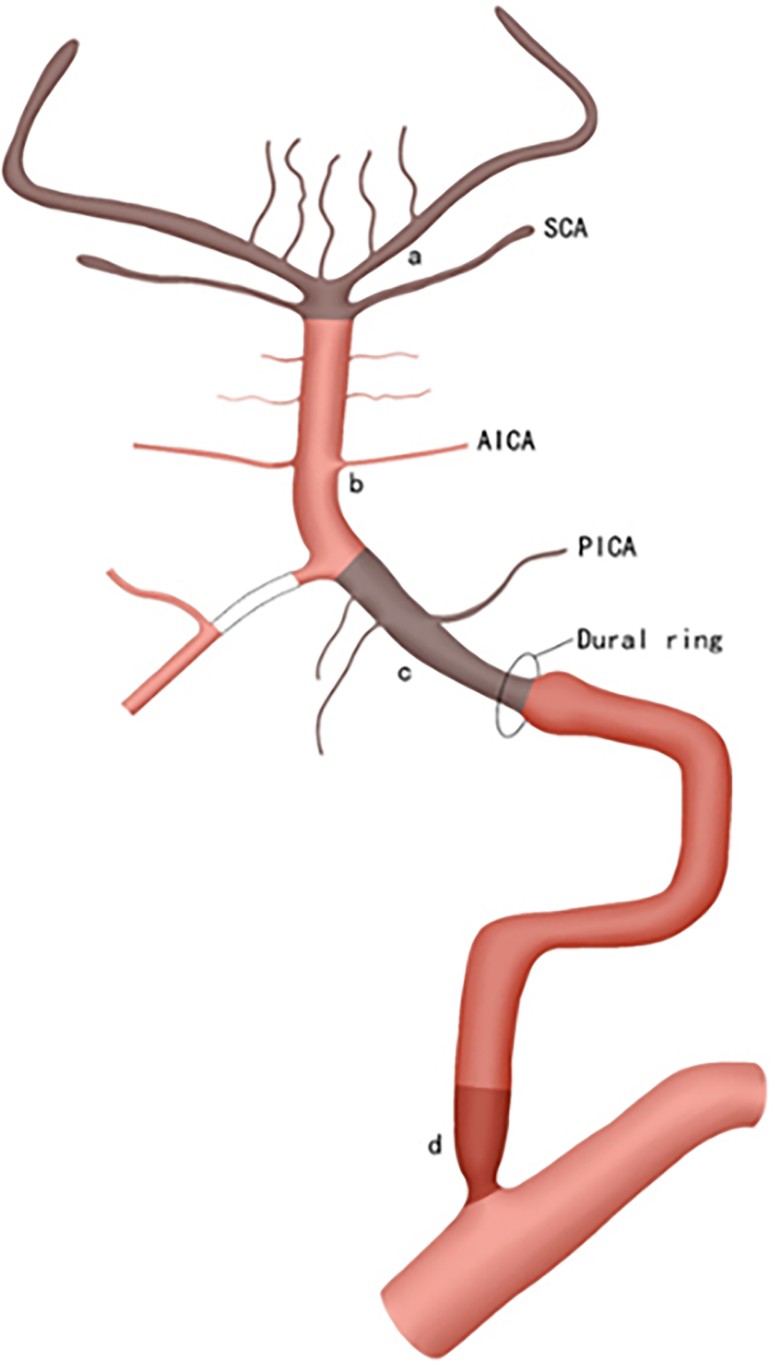

**Figure 2 The PCI is also categorized into four types based on the location of vascular occlusion.** a, b, c, and d; Type-a involves the distal BA, including the SCA; Type-b covers the region from the initial BA to, but excluding, the SCA; Type-c corresponds to the intracranial segment of the VA (*i.e.*, V4); Type-d encompasses the extracranial segment of the VA (*i.e.*, V1–V3), which is most commonly occluded at the origin of the VA.                           

Type-d is defined as occlusion of the extracranial VA, from V1 to V3. The DWI findings for type-d are either similar to those of type a or show large amounts of infarctions localized to one side of the posterior circulation. This is because large thrombi dislodged

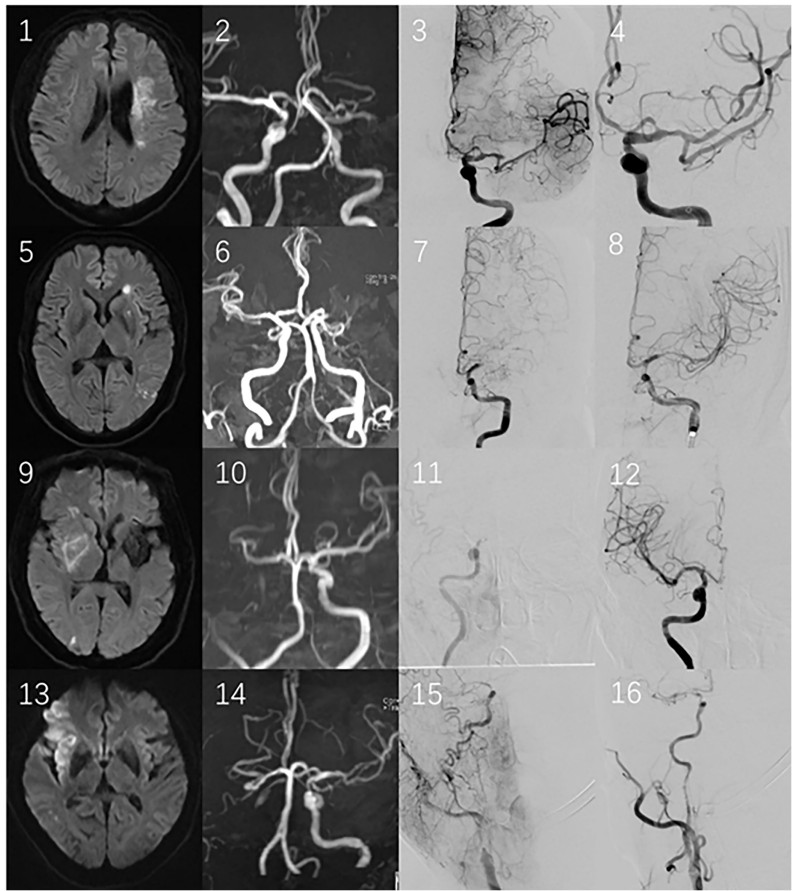

**Figure 3** **1–4 illustrate the first case, divided as type-A.** DWI (1) suggests acute cerebral infarction in the corona radiata and body of caudate nucleus, MRA (2) shows mid-distal occlusion of the left MCA, and EVT (3, 4) confirmed distal occlusion of M1; 5–8 belong to the second case, classified as type-B: DWI (5) displays lenticular nucleus infarction, MRA presents mid-distal occlusion of the left MCA, and EVT (7, 8) demonstrates middle occlusion of the left MCA; 9–12 pertain to the third case, falling into type-C: DWI (9) shows right PLIC infarction, MRA (10) indicates that the right ICA and MCA disappear, and EVT (11, 12) reveals the intracranial ICA occlusion. 13–16 are the fourth case, categorized as type-D: DWI (13) suggests the insular cortex and frontal lobe infarction, MRA (14) shows the right ICA and MCA disappear, and EVT (15, 16) certifies the occlusion of the initial part of right ICA, with the thrombi dislodging and leading to distal M1 embolism.

from the extracranial VA are influenced solely by the flow of the contralateral VA. MRA reveals that the VBA completely disappears, or the mid-distal BA and ipsilateral VA lose partial signal.

In the PCI classification, the vessel of type-a is so short that atherosclerosis or dissection is not considered to occur in this section solely.

**Etiologic classification:**

Etiologic classification, which is divided into three types (1–3), is based on the pathogenesis of AIS. The patients' medical history (including hypertension, diabetes, cardiopathy, tumor, smoking and so on) and electrocardiogram are analyzed to determine the pathogenesis of AIS. Patients with dissecting stroke often present with the typical characteristic of tearing pain in the head and neck, and patients with this characteristic

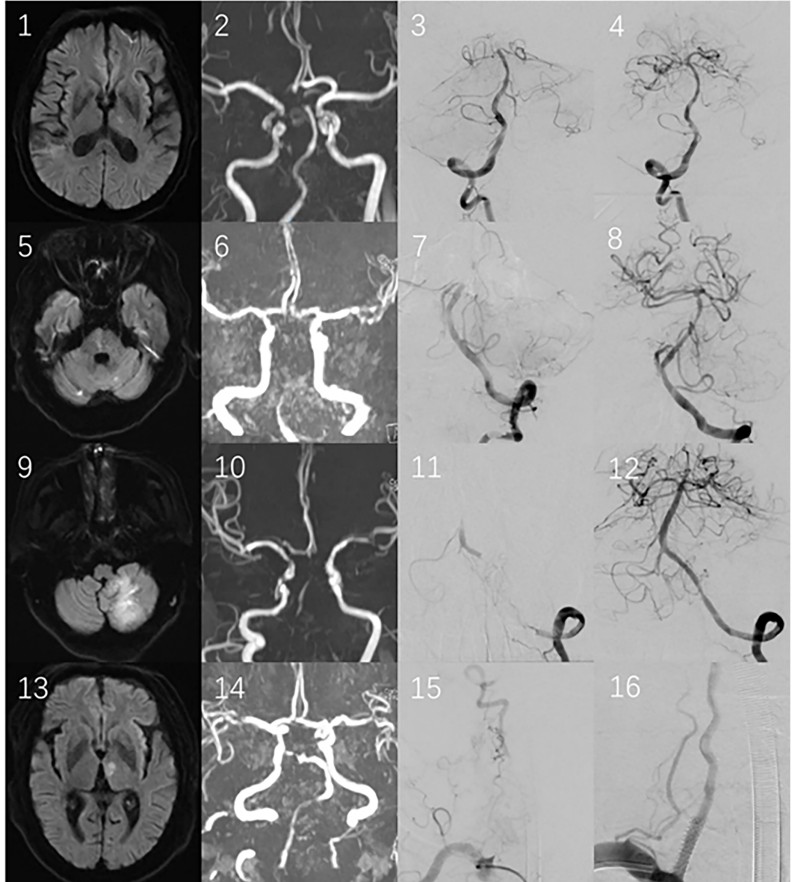

**Figure 4** **1–4 illustrate the first case, divided into type.** DWI (1) suggests acute cerebral infarction in the thalamus, MRA (2) shows occlusion at the top of the BA, which confirmed by EVT (3, 4); 5–8 belong to the second case, classified as type-b: DWI (5) displays pons infarction, MRA presents the BA and distal VA occlusion, and EVT (7, 8) demonstrates middle occlusion of the BA; 9–12 pertain to the third case, falling into type-c: DWI (9) shows the left cerebellar hemisphere infarction, MRA (10) indicates that the BA and VA disappear, and EVT (11, 12) reveals the intracranial VA occlusion. 13–16 are the fourth case, categorized as type-d: DWI (13) suggests thalamus infarction, MRA (14) shows occlusion at the top of the BA and partial signal loss in the VA. EVT (15, 16) certifies the occlusion of the initial part of the right VA, with the thrombi dislodging and leading to embolism at the top of the BA.

should be given priority consideration for type 3. Among stroke patients exhibiting atrial fibrillation, atrial fibrillation can account for approximately 79% of the causes of stroke. Thus, when multiple determinants are present at the same time, the priority of etiologic classification should be type 3, then 2, and then 1.

Type-1 is defined as large artery atherosclerosis (LAA), with or without thrombosis. Hypertension, diabetes, hyperlipidemia and smoking are the major risk factors for LAA; Patients who are over 40 years old but deny having any illness can also be classified into this type.

Type-2 is defined as thrombotic stroke. Cardiac disease, such as atrial fibrillation, atrial flutter and mitral valvular disease, is the most common risk factor causing thrombotic stroke. The other risk factors that have potential evidence for embolism and result in

**Table 2 The formula for calculating Youden's index and Kappa index.**

| Screening test | Gold standard | | Total | Sensitivity (%) | Specificity (%) | Youden's index | Kappa |
|---|---|---|---|---|---|---|---|
| | Positive | Negative | | | | | |
| Positive | A | B | $R_1$ | $A/N_1$ | $D/N_2$ | $A/N_1 + D/N_2 - 1$ | $N(A+D) - (R_1 N_1 + R_2 N_2)/N^2 - (R_1 N_1 + R_2 N_2)$ |
| Negative | C | D | $R_2$ | | | | |
| Total | $N_1$ | $N_2$ | N | | | | |

hypercoagulable state (HCS) include malignancy, bacteremia, septicemia, heavy use of hormones and renal failure. The patients who are younger than 40 years and deny any illness can fall into this type too, as well as acute multiple infarctions involving bilateral anterior and/or anterior and posterior circulations;

Type-3 is defined as vascular dissection, with or without thrombosis. Patients usually have a history of head-and-neck tearing pain.

### General description of the treatment procedure

AIS patients with LVO diagnosed through imaging and clinical symptoms, should be immediately sent to the green channel procedure for AIS. Then, diagnostic cerebral angiography is completed under local or general anesthesia, to fully evaluate collateral compensation and the position of cerebral vascular occlusion. For patients who need EVT, the neurointerventionist should choose the appropriate treatment measures according to the location of LVO, the pathogenesis and the individual experience. The common EVT procedures include mechanical thrombectomy, thrombus aspiration, intra-arterial thrombolysis, percutaneous transluminal angioplasty, *etc*. The patients are admitted to the general ward or ICU for further treatment after operation.

### Statistical analysis

The data were analyzed by SPSS 20.0 software. All normally distributed continuous quantitative variables were expressed as $\bar{x} \pm s$, non-normal distribution variables as $M (Q_1, Q_3)$. To assess the validity and reliability, each type of the imaging-etiologic classification was respectively compared to the gold standard intra-operative finding of EVT using a screening test. The closer the Youden's index approaches one, the higher is its validity, and the closer the Kappa index approaches one, the better is its reliability. The formulas for calculating the Youden's Index and Kappa are presented in Table 2.

Based on the methods described, we analyzed the data as follows.

## RESULTS

In our analysis of preoperative MRI, medical history, and intra-operative findings for 184 consecutive subjects with AIS, we found ACI represented 146 cases, accounting for 79.3% (146/184) and PCI 38 cases, accounting for 20.7% (38/184). Using the intra-operative variables of EVT as gold standard, the overall positive rate of the imaging classification meeting the gold standard was 93.5% (172/184), the etiologic classification was 94.6% (174/184), and the imaging-etiologic classification was 90.2% (166/184).

In this study, all vascular dissections occurred in the extracranial segments of the ICA and VA; It was observed that type-b only had atherosclerotic occlusion, while type-c and type-d were also primarily atherosclerosis-based, accounting for 76.5% (13/17) and 85.7% (6/7) respectively; Type-d was totally located in V1 or V1+V2 segments; In type-A, thromboembolism was more common, accounting for 62.5% (20/32). However, the proportions of atherosclerotic and thrombotic occlusion in type-B were nearly the same; Thrombotic occlusion was dominant in type-C, accounting for 75.9% (22/29); Type-D was predominantly caused by atherosclerosis, accounting for 65.8% (25/38), and mainly involved the initial and cervical ICA (that is C1), accounting for 78.9% (30/38).

Each type of the imaging-etiologic classification was evaluated separately using a screening test with the gold standard, and the results demonstrated its excellent validity (Sensitivity ≥ 75%, Specificity ≥ 90% and Youden's index ≥ 0.75) and reliability (Kappa ≥ 0.80) (Table 3).

## DISCUSSION

AIS stands as one of the numerous public health issues that plague the world, frequently contributing to death and disability. The TOAST classification system, a categorization framework used to evaluate stroke types and determine their underlying causes, categorizes strokes into five main categories: large artery atherosclerosis, cardioembolic stroke, small artery occlusion, strokes of other determined etiology, and strokes of undetermined etiology. Presently, classification systems like TOAST, rooted in etiology and pathogenesis, offer a theoretical basis for selecting distinct treatment protocols and medications in clinical practice. However, their directive significance for endovascular treatments remains somewhat limited. The newly conceived classification system, with a primary emphasis on evaluating large vessel occlusive strokes, endeavors to pinpoint the location and etiology of vessel occlusion, thereby facilitating targeted thrombectomy procedures. Nowadays, EVT for AIS patients with LVO is the recommended first-line treatment (*Powers et al., 2018*). In our center, MRI is the routine examination item for assessing AIS patients who have no contraindication to MRI and can cooperate during the examination. MRI requires at least six sequences, including T2-weighted, FLAIR, DWI, ADC, TOF-MRA and SWI. Changes in DWI and ADC can be identified even within minutes of stroke onset, making them more sensitive than CT in the first 6 h from onset (*Nael et al., 2014*). In the development of parenchymal edema, however, FLAIR changes may not be apparent until several hours from onset. This DWI/ADC-FLAIR mismatch has been utilized to estimate the time of stroke onset, and to assess thrombolysis or thrombectomy in patients within the unwitnessed or extended time window (*Thomalla et al., 2011*). The slow flow in arteries often results in complete loss of signal in TOF-MRA, which demonstrates intracranial vessels without using contrast agent. Hence, TOF-MRA may underestimate flow in distal branches, overestimate arterial stenosis or occlusion severity, and fail to fully estimate vascular status (*Boujan et al., 2018*). Although CTA and contrast-enhanced MRA can show occlusive vessels more clearly, they are not routine examination of AIS in our center due to objective reasons. Therefore, it is often necessary to make treatment decisions after MRI/CT examination in order to shorten the time before

**Table 3 The screening test results of imaging-etiologic classification.**

| | Type | Screening test | Gold standard | | Total | Sensitivity (%) | Specificity (%) | Youden's index | Kappa |
|---|---|---|---|---|---|---|---|---|---|
| | | | Positive | Negative | | | | | |
| ACI | A | Positive | 32 | 1 | 33 | 100 | 99.1 | 0.99 | 0.98 |
| | | Negative | 0 | 113 | 113 | | | | |
| | | Total | 32 | 114 | 146 | | | | |
| | B | Positive | 45 | 2 | 47 | 95.7 | 98.0 | 0.94 | 0.94 |
| | | Negative | 2 | 97 | 99 | | | | |
| | | Total | 47 | 99 | 146 | | | | |
| | C | Positive | 25 | 5 | 30 | 86.2 | 95.7 | 0.82 | 0.81 |
| | | Negative | 4 | 112 | 116 | | | | |
| | | Total | 29 | 117 | 146 | | | | |
| | D | Positive | 34 | 2 | 36 | 89.5 | 98.1 | 0.88 | 0.89 |
| | | Negative | 4 | 106 | 110 | | | | |
| | | Total | 38 | 108 | 146 | | | | |
| PCI | a | Positive | 5 | 1 | 6 | 83.3 | 96.9 | 0.80 | 0.80 |
| | | Negative | 1 | 31 | 32 | | | | |
| | | Total | 6 | 32 | 38 | | | | |
| | b | Positive | 7 | 0 | 7 | 87.5 | 100 | 0.88 | 0.92 |
| | | Negative | 1 | 30 | 31 | | | | |
| | | Total | 8 | 30 | 38 | | | | |
| | c | Positive | 17 | 0 | 17 | 100 | 100 | 1 | 1 |
| | | Negative | 0 | 21 | 21 | | | | |
| | | Total | 17 | 21 | 38 | | | | |
| | d | Positive | 7 | 1 | 8 | 100 | 96.8 | 0.97 | 0.92 |
| | | Negative | 0 | 30 | 30 | | | | |
| | | Total | 7 | 31 | 38 | | | | |
| Etiology | 1 | Positive | 93 | 9 | 102 | 98.9 | 90.0 | 0.89 | 0.89 |
| | | Negative | 1 | 81 | 82 | | | | |
| | | Total | 94 | 90 | 184 | | | | |
| | 2 | Positive | 78 | 1 | 79 | 90.7 | 99.0 | 0.90 | 0.90 |
| | | Negative | 8 | 97 | 105 | | | | |
| | | Total | 86 | 98 | 184 | | | | |
| | 3 | Positive | 3 | 0 | 3 | 75 | 100 | 0.75 | 0.85 |
| | | Negative | 1 | 180 | 181 | | | | |
| | | Total | 4 | 180 | 184 | | | | |

EVT. Generally, we use MRI to screen the candidates for thrombolysis, thrombectomy, craniotomy or conservative treatment, rather than to reveal the mechanism and precise position of vascular occlusion, which can only be obtained from procedures of EVT. This potentially increases the risk of EVT, extends the recanalization time and even causes serious consequence by choosing an improper EVT program. For AIS patients with LVO, the shorter the vascular recanalization time, the greater the benefit. We proposed a new

imaging-etiologic classification based on MRI and pathogenesis to locate the vascular occlusion and reveal its pathogenesis as quickly as possible. This classification can guide neurointerventionists in choosing the appropriate EVT program and shortening the vascular recanalization time.

In the new system, our imaging-etiologic classification achieved an overall positive rate of 90.2% (166/184) when compared to the gold standard. The screening test of each type demonstrated excellent validity and reliability: Sensitivity ≥ 75%, Specificity ≥ 90%, Youden's index ≥ 0.75 and Kappa ≥ 0.80. When the Youden index is ≥0.75, it indicates that the diagnostic test achieves a good balance between sensitivity and specificity, enabling accurate identification while effectively reducing misdiagnoses and missed diagnoses, thereby improving the accuracy and reliability of the diagnosis. When Kappa is ≥0.80, it signifies a very high level of agreement among raters, and their judgment results can be considered highly reliable. In this study, we found ACI accounted for 146, 79.3% (146/184) and PCI accounted for 38, 20.7% (38/184), that was consistent with the results of other studies in China. All of the vascular dissection occurred in the extracranial sections of the ICA and VA, which were the most predisposed sites. Additionally, thromboembolism frequently manifested in vascular bifurcations and Coarctation, as evident from our statistical data. Remarkably, throughout the study period, we did not encounter any adverse events associated with the imaging procedures or the endovascular treatments administered. These reassuring data underscore the credibility and suitability of the newly proposed imaging-etiologic classification system for clinical applications.

However, the real situation of AIS is occasionally complex, with occurrences such as thrombus migration and tandem occlusion. In order to fully uncover all the major occlusions, we must not confine ourselves solely to subclassifying the primary occlusion. For example, fragments of plaques or thrombi originating from the initial part of the ICA may dislodge and travel to distal branch, resulting in M1 embolism. In such cases, relying solely on type D-1 does not adequately capture the entire process, so we give D-1→A as a supplement; Similarly, Cardiac-origin thrombi may dislodge and travel to the distal ICA and proximal MCA, leading to a combined ICA and M1 embolism. Here, we give C/B-2 as a supplement to C-2. In the statistical analysis of data, these two scenarios are counted as D-1 and C-2 respectively, with other similar cases being treated analogously.

Vasculopathy in AIS is a intricate and dynamic process, and no relatively simple grading can accurately classify all the variations of such complex vascular lesions. For example, Cardiac-originated thrombi lead to occlusion of all intracranial and extracranial segments of the ICA, which will be wrongly classified as C-2 instead of D-2 (D/C-2) according to the imaging-etiologic classification system. The D-2 (D/C-2) typing indicates a higher thrombus load, which affects the vascular recanalization time and patients' prognosis. The anatomical variants, collateral circulation, and thrombus migration can make the interpretation of DWI lesions very challenging. In this classification system, the accuracy and consistency are the highest for type A and type c, whereas they are relatively low for type C and type a. This is primarily due to the fact that a larger thrombus load and thrombus displacement can easily lead to misdiagnosis. However, currently, there is a lack of rapid examination methods to accurately determine the thrombus load in emergency

situations. In the future, we can explore quick thrombus load examination methods to improve the accuracy of this classification. In spite of occasional misclassifications of the assigned grade, we still believe that the rapidity and simplicity of the classification system have great advantages. To take into consideration every possible anatomic and etiologic permutation of AIS, the system would have to be overly complex, making it impractical. This classification system which has demonstrated its validity and reliability, is simple, easy to learn, and easy to apply.

Some of our own suggestions for applying this classification system are as follows: 1, For large artery atherosclerosis, we propose to predict the thrombus load using the microcatheter "First-Pass Effect" first (*Yi et al., 2019*), then carry out mechanical thrombectomy with a stent to reduce the thrombus load. If there is higher residual vascular stenosis, perform angioplasty finally; 2, If it is defined as type-2, the thrombotic stroke, we prefer direct aspiration first-pass thrombectomy (ADAPT) and solitaire with the intention for thrombectomy (SWIM), which have been supported by many studies (*Lapergue et al., 2021, 2017; Turk et al., 2019*); 3, Unless the vascular dissection affects hemodynamics, we tend to select conservative treatment over EVT; Reducing the thrombus load and finding the true lumen as soon as possible are the top priorities of EVT for type-3; 4, For strokes identified as type-A, B, a, and b, ensure an adequate length and preferable diameter of the intermediate catheter (>115 cm, ≤5F); 5, As type-C is mainly thrombotic infarction, we recommend a large-diameter intermediate catheter (6F) to facilitate thrombus aspiration; 6, type-D is predominantly caused by atherosclerosis and often involves thrombus migration. Therefore, the balloon assisted track (BAT) technique or using a balloon guiding tube is a good choice; For cases with tandem occlusion due to thrombus migration, it is controversial whether to preferentially treat extracranial or intracranial lesions (*Wilson et al., 2018*); In our center, we primarily rely on the individual experience of the doctors.

As discussed above, the two parts of this classification are not independent of each other but require comprehensive consideration to assist the neurointerventionist in determining the main locations and etiologies of vascular occlusion and selecting the most appropriate solution.

## Study limitations

The main limitations of our study are as follows: 1, This study is not a randomized controlled trial; 2, The sample size of some types within this classification is small, and a larger sample size is needed for verification in the future studies; The uneven distribution of sample sizes among various categories and the existence of sample bias may lead to biased results, thereby limiting the stability of the research findings; 3, This study does not consider the factor of thrombosis load, which has an impact on patients' prognosis and the difficulty of EVT. However, we can gain a preliminary impression when interpreting DWI, which can be combined with this classification to jointly guide EVT; 4, There is a lack of prospective study on vascular recanalization rate, vascular recanalization time, complication rate and prognosis; In the future, we will design a randomized controlled study to evaluate the impact of the new classification system on enhancing the

recanalization rate of thrombus retrieval, shortening the recanalization time, reducing the complication rate, and improving patient prognosis.

## CONCLUSIONS

Our pilot study results show that our proposed imaging-etiologic classification can simply, rapidly and effectively reveal the location of vascular occlusion, the pathogenesis of AIS with LVO, and be used by interventionists to guide the choice of the appropriate EVT program.

### Funding

This work was supported by the Henan scientific and technological research projects (NO.232102521024). The funders had no role in study design, data collection and analysis, decision to publish, or preparation of the manuscript.

### Grant Disclosures

The following grant information was disclosed by the authors:
Henan scientific and technological research projects: 232102521024.

### Competing Interests

The authors declare that they have no competing interests.

### Author Contributions

- Hao Li conceived and designed the experiments, performed the experiments, analyzed the data, prepared figures and/or tables, authored or reviewed drafts of the article, and approved the final draft.
- Zhaoshuo Li conceived and designed the experiments, performed the experiments, analyzed the data, prepared figures and/or tables, authored or reviewed drafts of the article, and approved the final draft.
- Jinchao Xia conceived and designed the experiments, performed the experiments, analyzed the data, prepared figures and/or tables, authored or reviewed drafts of the article, and approved the final draft.
- Lijun Shen conceived and designed the experiments, performed the experiments, analyzed the data, prepared figures and/or tables, authored or reviewed drafts of the article, and approved the final draft.
- Guangming Duan conceived and designed the experiments, performed the experiments, analyzed the data, prepared figures and/or tables, authored or reviewed drafts of the article, and approved the final draft.
- Ziliang Wang conceived and designed the experiments, performed the experiments, analyzed the data, prepared figures and/or tables, authored or reviewed drafts of the article, and approved the final draft.

## Human Ethics

The following information was supplied relating to ethical approvals (*i.e.*, approving body and any reference numbers):

The Ethics Committee of Henan Provincial People's Hospital approval to carry out the study within its facilities.

## Data Availability

The raw measurements are available in the Supplemental Files.

## Supplemental Information

Supplemental information for this article can be found online at http://dx.doi.org/10.7717/peerj.18342#supplemental-information.

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
