# Peer review of "A retrospective analysis of a newly proposed imaging-etiologic classification for acute ischemic stroke with large vascular occlusion based on MRI and pathogenesis"

_PeerJ, doi:10.7717/peerj.18342_

## Round 0.1 · original submission · Major Revisions

Please response to the reviewers comments, point by point

·

Basic reporting

Suggest the author to improve their English writing skills. For example:
01. Enhance Clarity with Active Voice: The manuscript primarily uses passive voice, which is common in scientific writing. However, in some instances, switching to active voice can enhance clarity and directness. For example, instead of "The overall positive rate of imaging-etiologic classification meeting the gold standard was 90.2%," consider "Our imaging-etiologic classification achieved an overall positive rate of 90.2% when compared to the gold standard."
02. Refine Sentence Structure for Readability: Some sentences in the manuscript are quite long and complex, which can affect readability. Breaking these into shorter, more digestible sentences can improve the flow. For example, the sentence "The classification is divided into two parts, the first part is imaging subclassification and the second part is etiologic subclassification (Table-1)." could be restructured as "Our classification consists of two main components: imaging subclassification and etiologic subclassification, as detailed in Table 1."
03. Use of Transitions for Cohesion: The manuscript could benefit from the use of transitional phrases to improve the cohesion between paragraphs and sections. This helps guide the reader through the logical flow of the argument or findings. For instance, after the "Methods" section, a transition like "Based on the methods described, we analyzed the data as follows..." could smoothly lead into the "Results" section.

Experimental design

1. How was the inter-rater reliability of the imaging-etiologic classification established among the independent neurointerventionists who assessed the MRIs?
2. Could the authors provide more details on the statistical methods used to calculate the Youden’s index and Kappa statistics? Were there any corrections for multiple comparisons?
3. Could the authors discuss potential limitations and biases in the study design that might affect the generalizability of the findings?

Validity of the findings

The author has provided a lot of necessary supplementary materials to support the integrity and authenticity of the research. However, there are some flaws in the informed consent form, such as the lack of information about the potential benefits or risks that subjects may receive in the experimental study, and the research plan does not have remedial measures for potential risk events. Such flaws can seriously affect the effectiveness and safety of the research.
Of course, the most important thing is to explain why the research started in January 2020, while the ethics committee's approval for the research was in June 2020. What is the reason for this situation?

Additional comments

In addition, the author also needs to address the following questions:
1. Could the authors elaborate on the specific aspects of this new classification system that differentiate it from existing models, such as the TOAST criteria?
2. What are the expected changes in clinical practice as a result of adopting this new classification system? How does it impact patient outcomes compared to the current standard of care?
3. Is there a plan for prospective validation of this classification system in a larger and more diverse patient population?
4. Were there any adverse events related to the imaging procedures or the endovascular treatments that were recorded during the study?
5. The manuscript mentions that the study does not consider the factor of thrombosis load. Could the authors explain why this factor was excluded and its potential impact on the results?
6. Are there specific guidelines or criteria provided so that other institutions can reproduce the classification system accurately?
7. Was there any assessment of the cost-effectiveness of implementing this new classification system compared to traditional methods?
8. What are the next steps following this pilot study? Are there any planned studies to address the limitations identified in this research?

Reviewer 2 ·

Basic reporting

The article generally meets the basic reporting standards with clear and professional English throughout. However, I suggest the authors provide more comprehensive background on existing AIS classification systems in the introduction to better contextualize their novel approach.

Experimental design

The experimental design is generally sound and appropriate for addressing the research question. The authors have conducted a rigorous investigation using relevant imaging techniques and statistical analyses to validate their proposed classification system. However, I suggest providing more detailed information on the MRI protocols and assessment procedures to enhance reproducibility.

Validity of the findings

The findings presented in this study appear to be valid and well-supported by the data. The authors have provided a comprehensive statistical analysis that demonstrates the robustness of their proposed classification system. However, I recommend that the authors further discuss the potential limitations of their approach and consider addressing any outliers or unexpected results in more detail to strengthen the overall validity of their conclusions.

Additional comments

(1)I suggest the authors provide a detailed explanation of the rationale behind their inclusion and exclusion criteria.
(2)It would be beneficial if the authors could elaborate on the specific MRI examination procedures and parameter settings.
(3)Regarding the development of the imaging-etiologic classification system, I recommend that the authors clarify whether this was based on expert consensus or supported by existing literature.
(4)The authors mention using screening tests to evaluate the validity and reliability of the classification system. I suggest providing a more detailed description of the specific methods and criteria used in these screening tests to allow readers to better assess the robustness of the evaluation process.
(5)I recommend that the authors explain their rationale for choosing Youden's index ≥0.75 and Kappa ≥0.80 as the standards for excellent validity and reliability.
(6)It would be helpful if the authors could describe how consensus was reached between assessors, particularly whether any specific methods were employed to resolve disagreements.
(7)Regarding the three types in the etiological classification, I suggest the authors explain how the priority order among these types was determined.
(8)In the discussion, the authors mention special cases such as thrombus migration and tandem occlusion. I recommend further elaboration on how these situations are accommodated within the classification system to demonstrate its comprehensiveness.
(9)I suggest the authors explain their reasoning for dividing both anterior and posterior circulation into four subtypes each, rather than a different number of subtypes. This would help readers understand the structural logic of the classification system.
(10)It would be valuable if the authors could discuss the advantages and limitations of their classification system compared to existing systems like TOAST.
(11)Given that the overall concordance rate of the classification system reached 90.2%, with excellent validity and reliability for each type, I recommend a deeper analysis of these important results in the discussion. The authors could explore whether certain types had particularly high or low accuracy rates and discuss possible reasons. They could also elaborate on the implications of this high concordance rate for clinical practice and how the accuracy of the classification might be further improved.

---

## Round 0.2 · accepted · Accept

I think the manuscript could be accepted at the current version. Congratulations.

·

Basic reporting

no comment

Experimental design

no comment

Validity of the findings

no comment

Additional comments

no comment

Reviewer 2 ·

Basic reporting

The author has addressed my previous concerns thoroughly and made significant improvements to the manuscript based on the feedback provided. The revisions demonstrate a clear commitment to enhancing the quality and clarity of the work. Given these substantial improvements, I believe the manuscript now meets the standards required for publication in this journal. The revised version effectively contributes to the existing body of knowledge in the field. Therefore, I recommend acceptance of this paper.

Experimental design

The author has addressed my previous concerns thoroughly and made significant improvements to the manuscript based on the feedback provided. The revisions demonstrate a clear commitment to enhancing the quality and clarity of the work. Given these substantial improvements, I believe the manuscript now meets the standards required for publication in this journal. The revised version effectively contributes to the existing body of knowledge in the field. Therefore, I recommend acceptance of this paper.

Validity of the findings

The author has addressed my previous concerns thoroughly and made significant improvements to the manuscript based on the feedback provided. The revisions demonstrate a clear commitment to enhancing the quality and clarity of the work. Given these substantial improvements, I believe the manuscript now meets the standards required for publication in this journal. The revised version effectively contributes to the existing body of knowledge in the field. Therefore, I recommend acceptance of this paper.

Additional comments

The author has addressed my previous concerns thoroughly and made significant improvements to the manuscript based on the feedback provided. The revisions demonstrate a clear commitment to enhancing the quality and clarity of the work. Given these substantial improvements, I believe the manuscript now meets the standards required for publication in this journal. The revised version effectively contributes to the existing body of knowledge in the field. Therefore, I recommend acceptance of this paper.